# Transport in the Brain Extracellular Space: Diffusion, but Which Kind?

**DOI:** 10.3390/ijms232012401

**Published:** 2022-10-17

**Authors:** Eugene B. Postnikov, Anastasia I. Lavrova, Dmitry E. Postnov

**Affiliations:** 1Theoretical Physics Department, Kursk State University, Radishcheva St. 33, 305000 Kursk, Russia; 2Institute of Physics, Saratov State University, Astrakhanskaya Str. 83, 410012 Saratov, Russia; 3Saint-Petersburg State Research Institute of Phthisiopulmonology, Ligovskiy Ave., 2-4, 194064 Saint Petersburg, Russia; 4Medical Faculty, Saint-Petersburg State University, Universitetskaya Emb. 7/9, 199034 Saint Petersburg, Russia

**Keywords:** brain fluids, extracellular space, diffusion

## Abstract

The mechanisms of transport of substances in the brain parenchyma have been a hot topic in scientific discussion in the past decade. This discussion was triggered by the proposed glymphatic hypothesis, which assumes a directed flow of cerebral fluid within the parenchyma, in contrast to the previous notion that diffusion is the main mechanism. However, when discussing the issue of “diffusion or non-diffusion”, much less attention was given to the question that diffusion itself can have a different character. In our opinion, some of the recently published results do not fit into the traditional understanding of diffusion. In this regard, we outline the relevant new theoretical approaches on transport processes in complex random media such as concepts of diffusive diffusivity and time-dependent homogenization, which expands the understanding of the forms of transport of substances based on diffusion.

## 1. Introduction

The question of the nature and mechanisms of the transfer of substances in the intercellular space of the brain parenchyma is very important, both for finding ways to deliver drugs and for understanding how the brain gets rid of harmful metabolites. By the first decade of our century, there was a consensus among experts that, under special conditions, cerebrospinal fluid can penetrate deep into the parenchyma [1,2,3], and then enter the bloodstream [4,5]. The driving force of this movement was considered to be a slight hydrostatic pressure created by the secretion of fluid through the blood-brain barrier [6] or the movements of the walls of the arteries [7].

Since 2012, these issues have been actively discussed in connection with the so-called glymphatic hypothesis proposed in [8,9,10,11]. According to this hypothesis, the transport of substances through the parenchyma is provided by a directed flow of cerebral fluid maintained by a pressure gradient. However, a number of researchers do not share this point of view and argue in favor of the diffusion mechanism.

As far as the authors know, at the moment there is no experimental evidence of the presence of a directed fluid flow through the parenchyma under normal physiological conditions. In a number of works [12,13,14,15,16,17] the thesis is supported that diffusion is the main and sufficient mechanism for the delivery of substances in the parenchyma. In [14], the purely diffusion nature of the transport of substances in the parenchyma was confirmed experimentally. Theoretical estimates predict that flows generated by achievable hydrostatic pressure drops will be negligible compared to the diffusion effect [13,14,15,16,18]. A recent review [19] analyzed the current situation and formulated a list of problematic points of the glymphatic hypothesis. At the same time, in [20], the authors present a number of arguments in favor of the glymphatic mechanism, somewhat clarifying (softening) the wording. In particular, they acknowledge that “whether advection is appreciable in the interstitial compartment remains an open question”.

Thus, there is general agreement that diffusion is, if not the exclusive, then the main mechanism of transport in the extracellular space (ECS) of the brain parenchyma. However, in the course of the discussion described above (diffusion or non-diffusion), the question of the characteristics of the diffusion process itself received much less attention.

The features of diffusion in the ECS of the brain parenchyma have been purposefully studied by Nicholson with collaborators over the past 20 years [21,22,23,24,25,26,27,28]. Typical properties and features of ECS as a medium for the propagation of molecules, including a complex shape, dead spaces (traps), etc., are revealed. With regard to the diffusion process itself, in these works, a line is drawn for taking into account all the features of the ECS by means of (i) the tortuosity of the medium, which describes the lengthening of the random walk path of molecules and (ii) the effective diffusion coefficient, which phenomenologically takes into account the slowdown of the real diffusion process in respect to the idealized process due to the above reasons.

At the same time, the fact that the diffusion process itself can have different properties is still beyond the attention of researchers. In recent years, new approaches to the theory of transport processes in complex random media, such as concepts of diffusive diffusivity and time-dependent homogenization, have been proposed and developed, which expands the understanding of the forms of transport of substances based on diffusion [29].

The purpose of this paper is to draw attention to this issue and to the fact that already published experimental data may have an alternative interpretation in terms of the so-called Brownian-yet-not-Gaussian diffusion. New theoretical models can be of real benefit in studies of molecular transport in the brain only if the experimental methods provide the necessary accuracy and spatiotemporal resolution. For this reason, in our work, we briefly discuss methods for studying diffusion using magnetic resonance imaging (MRI), as well as single particle tracking (SPT), which, in our opinion, are capable to provide relevant information.

## 2. ECS Transport Assessment

There are well-known and well-described approaches as studying the marker’s spread in the ECS as registering the diffusion process by means of fluorescent optical method and the real-time iontophoresis technique. On these topics, we refer readers to a number of already existing reviews. Among them, we can list the comprehensive description of the super-resolution fluorescent optical methods aimed in highlighting ECS structures in living brain [30] as well the respective topics in comprehensive review [31]. Respectively, to iontophoresis and its interpretation for the brain’s ECS structure, nobody would describe this topic better the inventor of the method, C. Nicholson. We cited most related his works in Introduction. Finally, it is worth noting the review [32], which highlights variety of methods adjusted to such carriers of markers’s clouds (flurescent, radiocontrast, etc.) as nanoparticles.

The present review is aimed to be focused on some non-trivial features, which may be overlooked within the conventional analysis of transport processes in the brain’s parenchyma. Thus, in the following subsections, we draw attention to two experimental techniques, which have potential to provide data of a special interest related to non-conventional kinds of diffusion.

### 2.1. MRI-Based Diffusion Studies

Magnetic resonance imaging (MRI) has certain advantages over other methods of studying transport processes in the brain due to the possibility to access deep layers and obtain 3D images than is either hard realisable or impossible by optical methods [33,34,35]. The particular problem of determining the apparent diffusion coefficient can be solved by applying the “pulsed gradient spin-echo” method, which utilizes a pair of narrow gradient pulses and relates the phase change of the responses to the phase change of an ensemble of spins due to the random transport of particles carrying them during the used diffusion time. As a result, the diffusion-weighed MRI (DW-MRI) provides information on different scales of tissues, from the structure-related diffusional features to the global mapping of organoids, see, e.g., for review, the Special Issue of the journal “NeuroImage” [36].

The conventional DW-MRI operates with proton spins of water molecules. Thus, the measured diffusion rate (as well as other transport coefficients) relates primarily to water transport, which coexists intracellularly and extracellularly, depends on the membrane permeability of the brain’s compartments, an action of water-transporting proteins, etc. All these factors complicate the assessment of diffusion specifically in ECS.

An improved method was proposed about a decade ago [37,38] based on an injection of a Gadolinium-based contrast agent (e.g., Gadolium-diethylenetriaminepentaacetic acid (Gd-DTPA), Gd-chelates such as Dotarem^®^, ultra-small Gd-enhanced super-paramagnetic iron oxide nanoparticles, or even emulsions incorporating some tenth thousands of Gd atoms), which do not permeate cellular membranes and spread mainly in the extracellular space. This spread is registered as a time sequence of T1-weighted images (i.e., using short Time-to-Echo and repetition time intervals) with the subsequent their conversion to concentration maps, respectively, to the initial image registered before introducing a Gd-based agent.

Among the recent works, one can note a set of measurements in vivo carried out with this method [39,40], which indicated not a perfect reconstruction of the contrast agents infused in animal’s brain. We will consider possible implications of these data within the context of non-standard diffusion further in Section 4.2.

### 2.2. Single Particle Tracking

Among modern methods which recently gain importance in the study of transport processes in complex media, a special place takes the single-particle tracking (SPT) because it allows direct accessing paths of marker particles in heterogeneous labyrinthine structures, structures with dead ends and traps, among multiple obstacles, etc. In particular, obtaining real detailed tracks provide an opportunity of mapping the actual topology of spaces available for transport and highly-localised quantification of transport-supporting properties. Since such media are especially typical for biophysical objects, the main developments of the SPT approach accompanied by the developing methods of super-resolution in optical microscopy is tightly related to biophysical problems, see reviews [41,42], which give a general overview of general physical principles and examples of case studies.

One of the useful features of SPT is a variety of usable markers that includes organic dyes (about 1–2 nm in the characteristic size) and fluorescent proteins (about 2–4 nm), quantum dots (about 5–8 nm before their biofunctionalization), or significantly bigger gold nanoparticles (about 400 nm). The variety of markers’ sizes and types of their interaction with the surrounding allows specifying accurately ranges transport gaps and taking into account interactions with cells’ membranes surrounding these gaps.

Respectively, to the exploration of the brain’s ECS, SPT methods gave reconstruction of the ECS’s topology and characteristic distances between cell providing transport channels [43,44,45,46,47]. Another usage is the SPT-based data is related to the fact of mutual interchangeability of studying mean-squared displacement (MSD) of random walker exploring either one long track or the evolution of the concentration distribution of a large ensemble of markers in the case of classic diffusion process (ergodicity). The MSD in this case directly gives the value of the diffusion coefficient. Thus, analysing tracks of single particles in different locations of the brain’s ECS, it is possible to get a map of the distribution of *local* diffusion coefficients, which as revealed widely varies across the ECS [45,47,48,49].

A very recent review [50] provides a comprehensive overview of methods based on the single particle tracking for the optical imaging-enabled diffusion measurement in the brain ECS, as well as technical details of the method’s implementation. A ground-breaking work [51] describes the proposed and realised methods for the quantification of distributions of spaces available for diffusive transport in brain’s ECS, local diffusivities and viscosities of the interstitial liquid. The method is based on the intraventricular injections of near-infrared luminescent single-walled carbon nanotubes (SWCNTs) in the brain of living rats with the subsequent preparing neocortex slices, where the transport of nanotubes was monitored.

The SPT method may have certain advantages for studying transport processes in the brain’s ECS since a single particle tracked during its random walk in the intercellular space directly highlights the shape and permeability of the latter. This opens two perspectives: (i) when one operates with diffusivities considered as almost constant in limited spacial regions but having different values in spatially separated places, one can plot a map of diffusivities [52]; (ii) when the mean-squared displacement of the walker exhibit anomalous (slower than linear) time dependence, exploration of single tracks allows characterising the particular type of this process distinguishing e.g., between waiting time distribution due to trapping in dead ends and walks in significantly highly inhomogeneous (say, fractal) structures, see [53]. In more detail, we will consider some particular features of results obtained via the SPT in Section 4.3.

## 3. Physics of Diffusion: What Can Be Searched for in Brain’s ECS?

From the physical point of view, the process of diffusive transport can be considered either macroscopically or microscopically. In many cases, these two approaches do not contradict each other. These cases belong to so-called ergodic processes: ergodicity means that there is no difference between averaging observable quantities over an ensemble of elements or over time averaging for one element observes sufficiently long time. The usual diffusion is an ergodic process, and one can characterize it with a single diffusion coefficient *D*, either typical for a medium or an effective one. It is possible to calculate the value of *D* from the mean-squared displacement (MSD) as MSD(t)=2NDt considering a single track in *N*-dimensional space
$$MSD_{t}\left(t\right)=\frac{1}{T-t}\int_{0}^{T-t}{\left|r\left(t^{\prime }+t\right)-r\left(t\right)\right|}^{2}dt^{\prime }.$$
where *t* is called the lag time and *T* is the length of positional time series r(t).

Respectively, when there is distribution of positions of random walkers belonging to an ensemble, the second moment of this distribution is $MSD_{e}\left(t\right)=\langle {\left|r(t)-r(0)\right|}^{2}\rangle$.

When the diffusivity (i.e., diffusion-related medium features) is spatially inhomogeneous, the relations above may be less trivial. However, if inhomogeneity is formed by patches with locally uniform properties, the relation between the local macroscopic diffusion coefficient and the value determined by the SPT is still valid. In this case, one simply needs to consider segmentation of the full track in shorter subintervals for analysis, see the more detailed discussion in [52].

If medium inhomogeneity has the form of some complex structure and the marker’s spread covers such an area, new specific features of the diffusion process can be observed, as recently revealed in modern physical considerations.

### 3.1. Gaussian (Fickian) Diffusion and Point-Source Paradigm

Macroscopically, one considers the flux q=−D∇n of the concentration *n* combined with the continuity equation $\partial_{t}=\nabla \cdot q$. For the constant diffusion coefficient *D* that gives the standard (Fickian) diffusion equation
(1)$$\frac{\partial n}{\partial t}=D\nabla^{2}n.$$

In an unbounded domain in a point parameterised by the vector of coordinates r, it has the solution as
(2)$$n\left(r\right)=\int n_{0}\left(r^{\prime }\right)\frac{e^{-}\frac{|r-r^{\prime }{|}^{2}}{4Dt}}{{\left(4\pi Dt\right)}^{N/2}}dr^{\prime }d\Omega \left(r^{\prime }\right),$$
where *N* is the dimension of space and $d\Omega (r^{\prime })$ is its volume element, $n_{0}\left(r^{\prime }\right)$ is the initial distribution of the spreading marker. In the case of a point source, when $n_{0}\left(r^{\prime }\right)=\delta \left(r^{\prime }\right)$, i.e., the initial condition is the Dirac delta-function, this solution reduces to the Gaussian function in the integrand of Equation (Equation 2). On this basis, the “point-source paradigm” was introduced by C. Nicholson [22] for studying transport processes in brain tissues. This method is based on the assessment of the structural properties of the extracellular space through the parameters of the Gaussian function fitted to the concentration distribution data.

Within this approach, an influence of the possible volume heterogeneities and properties of interstitial fluid in the brain’s extracellular space are counted with the concept of tortuosity [25], which is defined as the square root of the ratio of the diffusion coefficient of the same marker in a free medium to the considered effective diffusion coefficient and $\lambda =\sqrt{D_{free}/D}$.

Figure 1A,B illustrates this situation in terms of single particle trek. Note, the specific trek between points A and B may be shorter (faster) for obstacled medium (panel (b)) in respect to panel (a), but in average (over many particles)—it is longer (slower). The typical range of λ in ECS is about 1.4–1.7 when, say, a diluted agar gel is taken as the reference medium. Note that the tortuosity is a unified phenomenological quantity, which takes into account the free volume between cells, volume topology, cumulative effects of viscosity of the interstitial liquid, etc. [23,54].

Thus, recently it is the main method characterising transport parameters for different small and macromolecules in the brain parenchyma, see, e.g., the reviews [15,27,31].

It is also worth noting the recent active interest among specialists in material science to the experiments and modelling of transport processes in porous media considered a hindered diffusive spread characterized by the medium’s tortuosity. It was studied by the nanoparticle tracking inside of fibrous and porous media [55] or related to different expressions of the tortuosity as a function of the porosity [56]. These studies seem to be promising for brain-related problems since such media can play the role of a phantom mimicking the brain’s parenchyma tissue, the issue of the high recent demand for clarifying physical mechanisms of molecular and nanoparticles transport as highlighted in the recent reviews [57,58,59].

It should be pointed out that accessing diffusivity in the brain’s parenchyma not in specially prepared laboratory systems but realistic conditions of distributed infusion of marker substances, e.g., due to permeability of the blood-brain barrier, may make the point-source paradigm inapplicable. However, the usage of the full integral form of Equation (Equation 2) and fitting of registered concentration distributions to a set of trial solutions with different *D* with the choice of the best value, makes this problem of determining an effective diffusion coefficient solvable as shown in the work [60].

Note also that while completely impenetrable boundaries of obstacles, between which ECS transport occurs, are modelled by boundary conditions supplying Equation (Equation 1), an explicit spatial distribution of diffusivity within the flux-based approach leads to the form
(3)$$\frac{\partial n}{\partial t}=\nabla \cdot \left(D(r)\nabla n\right),$$
which allows also addressing average properties of complex fractal structures of some regions of the brain’s parenchyma, in particular, cerebellum, where anomalous (non-Gaussian) diffusion has been detected experimentally [61].

Within the point-source paradigm, one can consider the isotropic radius-dependent diffusion coefficient $D\left(r\right)=D_{\theta }r^{-\theta }$, which been substituted in Equation (Equation 3) results in the O’Shaughnessy-Procaccia equation [62] with solution describing anomalous diffusion (i.e., the second moment of the concentration’s distribution scales as $\sigma^{2}∼t^{2/(2+\theta )}$ on fractals with the scaling index θ (for the normal diffusion θ=0). The algorithm of image processing, which compares the experimentally-detected distribution of markers with trial solutions of the O’Shaughnessy-Procaccia equation allows quantitatively reveal the fractal dimension of the ECS in this case [63].

### 3.2. Non-Gaussianity in Diffusion

The highly irregular structure of the space, where the diffusion occurs, may result in types of diffusion, which are beyond the conventional diffusion process. Recent discoveries related to such and similar non-conventional kinds of diffusion in complex media are reviewed in [64]. Specifically, the authors list biophysical systems, where non-Gaussian diffusion is observed (e.g., the transport in viscoelastic intracellular environments and on the cells’ surface, in a non-homogeneous landscape hindering the random walk by the interactions of moving particles (“walkers”), for example, molecules or nanoparticles, with proteins, etc.), mathematical methods of quantifying non-Gaussianity and the basic mathematical approaches for modelling such a behaviour.

In this section, we address some of these findings from the microscopic point of view, i.e., describe diffusive spread in terms of dynamics of the spatial probability density function of a walker’s position associated with the concentration in the macroscopic consideration. In this case, one starts from the Langevin equation
(4)$$\dot{r}=\sqrt{2D(r)}\xi \left(t\right),$$
where ξ(t) is the uncorrelated Gaussian noise.

#### 3.2.1. Ito versus Hänggi–Klimontovich Interpretation

When D=const, the probability density function (the concentration distribution) satisfies Equation (Equation 1) and the whole picture in the macroscopic level is the same as described above. On the contrary, when D(r) is position-dependent, the correspondent Fokker-Planck equation depends on the interpretation of random walks realizations and has the general form
(5)$$\frac{\partial n}{\partial t}=\nabla \cdot \left[(1-\alpha )\nabla D(r)+D(r)\nabla \right]n,$$
where e.g., α=0 for Ito processes, when the diffusivity is taken in the random jump’s starting point, and α=1 for the Hänggi–Klimontovich interpretation when diffusivity in considered in the jump’s target point. Note, only the Hänggi–Klimontovich interpretation, i.e., the case of α=1, leads to the phenomenological second Fick’s law (Equation 3), while Ito’s case results in the equation
(6)$$\frac{\partial n}{\partial t}=\nabla^{2}\left[D(r)n\right],$$
with its solution not coinciding with Equation (Equation 2).

Remarkably, it was revealed that Equation (Equation 6) is the better choice to describe the diffusion in the gelatinous liquid [65]. This substance is similar to those, which are used as phantoms mimicking brain tissue [66,67,68]. Thus, it gives a hint for processing data obtained in experiments with the real biological ECS.

Another feature distinguishing between solutions of Equations (Equation 2) and (Equation 6) in a complex medium with random (but locally correlated) diffusivities was revealed in the work [69], see Figure 2: the case of Ito process results in a patchy concentration distribution in contrast to Hänggi–Klimontovich’s. When operating with data for distributions in real biological brain tissue (see e.g., figures in the work [39]), one would see certain patchiness too.

Thus, some attentional analysis, aimed at the distinction between instrumental noise and possible manifestations of features of the transport and ECS’s structure, makes sense. This may reveal biophysically relevant features of the extracellular space specificity because, as proven in the work [70] these two kinds of transport processes have clear different microscopic interpretations:

Ito’s model originates from the existence of *trap*, while Hänggi–Klimontovich’s originates from the existence of *barriers* for spreading walkers, see Figure 1, panels (C) and (D), respectively. Both these types of obstacles exist in the ECS: the typical traps affecting the diffusion in the brain’s extracellular space are dead ends [23] while barriers can be associated with the loss of diffusing material described by the kinetic interaction with ECS channels’ walls [26]. Thus, quantifying the type of concentration distribution can reveal the prevalence of obstacles of a particular kind. Moreover, their combination addresses the general form given by Equation (Equation 5) with 0<α<1. This also can provide important insights into the biophysical mechanisms of extracellular transport on a molecular level of resolution despite the fact that such microscopic interactions are not accessible for direct observations.

#### 3.2.2. Subdiffusion

Note, the Gaussian noise is not obligatory as a source of displacements in Equation (Equation 4). Another kind of random walk may lead to anomalous diffusion when the growth of the mean-squared displacement is scaled as $\sigma^{2}∼t^{\beta }$ with β≠1. Specifically, the case of viscoelastic media or/and the existence of traps leads to subdiffusion with β<1.

It should be pointed out that the same index of subdiffusion β can correspond to different microscopic physical mechanisms of transport, for example, the continuous time random walk originated from trapping for random time intervals, a viscoelastic character of the carrier fluid, which induces retardation of the motion due to the memory effects, geometric reasons such as a random walk in fractal structures as well as a combination of different origins. Methods which allow distinguishing between the mechanisms listed above are reviewed and explained in the work [53]. Comparative analysis of different algorithms for the classification of types of diffusive transport based on the individual tracks’ recordings and well as their segmentation can be found also in [71].

There is no yet clear experimental evidence of anomalous diffusion in the brain’s ECS (in contrast to intracellular transport and transport of water molecules [45,72,73]), although some primary recent observations can be noted, e.g., [74]. For this reason, we simply refer to some classic reviews of anomalous diffusion considered from the random walk point of view [75,76], including the special attention to the processes, which occur in complex (including labyrinthine) environments [77], without the detailed consideration of such mechanisms.

#### 3.2.3. Brownian Yet Non-Gaussian Diffusion: Diffusing Diffusivity and Quenched Spacial Heterogeneity

As a more valuable and prospective idea for the area of study of transport processes in the brain’s ECS, there is another phenomenon that was discovered not long ago: the Brownian yet non-Gaussian diffusion (BnG) [78]. Its key feature is that even if the macroscopic MSD follows the regularity typical for the usual Brownian motion $\sigma^{2}∼t$, the probability distribution function (equivalent to the registered concentration distribution detectable in experiments with the initial point source) follows not the Gaussian but the Laplace distribution
(7)$$n\left(r,t\right)∼\frac{1}{\langle D\rangle t^{N/2}}\exp\left(-\frac{\left|r\right|}{\langle D\rangle t^{1/2}}\right)$$
with the effective diffusivity 〈D〉.

Note that this phenomenon does not originate from a special kind of unique random motion but rather from the complexity of a medium that leads to the case of superstatistics as pioneered by Beck & Cohen [79]; see also a review of the recent state of the art is given by R. Metzler [29]. The essence of this idea is that walkers move inside a complex media with local properties determining the diffusive spread and significantly fluctuating either in space or in time. This results in operating with an ensemble with a distribution of mobilities, the Laplace distribution emerges due to averaging
(8)$$n\left(r,t\right)=\int_{0}^{\infty }p\left(D\right)\frac{e^{-}\frac{|r-r^{\prime }{|}^{2}}{4Dt}}{{\left(4\pi Dt\right)}^{N/2}}dD$$
with local diffusivities *D* satisfying some appropriate probability density function p(D).

One of the possible mechanisms of randomness in *D* is the model of *diffusing diffusivity* proposed in [80] and analysed in details in [81]. Within this concept, the diffusion coefficient exhibits slow temporal fluctuations. This implies that the evolution of the respective distribution of diffusivities p(D(t)) itself satisfies some kind of the diffusion (or advection-diffusion) equation that explains the terms “diffusing diffusivity”. The consideration of the effective diffusion coefficient as a stochastic process itself matches some examples of biological media [78,82].

Another variant considered in detail in the work [69] refers to a random locally correlated static landscape, i.e., when diffusion coefficients are slowly varying in space (quenched disorder) but not in time. The principal difference from the case of diffusing diffusivity is that when a random walker visits the same place of a heterogeneous medium again, it crosses the region with the same diffusion coefficient as at previous encounters of the same spatial location. At the same time, by recording the instant diffusivities at each time moment along a single trajectory, one will see the fluctuating time series and the BnG-type behaviour at short time scales is evolving too. At large times, the trajectory covers a significant part of space formed by the random but stationary landscape and one can apply the theory of homogenization to get an asymptotic effective diffusion coefficient for the resulting normal diffusion. Simulations revealed such non-Gaussianity in compartmentalized media [83], which mimic micro-organoids (like cells) separated by inter-organoid (intercellular) spaces.

## 4. Evidence of Non-Classical Diffusion in Brain

### 4.1. Where Can It Be Caught

It should be pointed out that the majority of works representing analysis of transport processes assume *a priory* that the registered distribution of spreading markers should be fitted by an appropriate Gaussian function. However, as it is discussed above, diffusion in a complex medium can exhibit a wider variety of types. Thus, it is worth reconsidering with a fresh eye images of the markers’ spread in the brain tissue published in a number of experimental works.

In particular, one can note a difference between cases when the Gaussian function fits the marker distribution in the reference substance—agarose—and in the cortex provided in one of the groundbreaking early works [24]. For the initial distribution, e.g., narrow-spread shortly after injection, the Gaussian function fits perfectly concentration distributions in agarose. However, there are visible deviations in the case of the cortex.

Although it is hard to judge about the significance of such small deviations in the cited and several subsequent works based on the iontophoretic and fluorescence-based experiments, e.g., [84], the experiments, which used MRI technique, e.g., [39,40], demonstrates the significant deviations from Gaussianity.

This opens the question, of whether simple diffusion with a unique effective diffusion coefficient is always adequate to the description of the transport processes in the brain’s ECS, and maybe one needs to revisit these experiments from the point of view of modern views on the Brownian yet non-Gaussian diffusion. An additional argument for such reconsideration is provided by the single-particle tracking experiments, which indicate that the local diffusivities in the ECS are wide-range distributed [48], see also the review [50]. However, this is the key premise of the BnG models discussed above. In addition, it should be pointed out that the single-particle tracking experiments may reveal specific features, which are “masked” in the case of large ensembles of markers [71].

### 4.2. MRI Results: Brownian Yet Non-Gaussian Diffusion?

A more detailed exploration of results obtained via this approach can be found in the works [39,40].

Note that authors of all these investigations *a priory* assumed the Gaussian distribution of the marker’s concentration as following the from the normal diffusion process within their analysis of results of such measurements. At the same time, even the visual exploration of the plots given in the cited papers induces some questions as to their relevance. In particular, the fitting Gaussian functions reproduce mainly the intermediate part of experimental distributions exhibiting significant deviations in the central part and the tails. The central part looks like a peak, which disappears very slow in comparison with the more distant parts of the distribution. Simultaneously, the tails deviate from the Gaussian fit for small times of the process and the correspondence starts to be better with time (in particular, authors of Ref. [40] use two Gaussians for fitting the central and the tail parts of distributions, for this reason).

However, these features resemble recently analysed cases of the diffusion process in a random environment [69,85] when one can approximate the diffusion coefficient by some averaged (homogenised) value only asymptotically. Otherwise, the process belongs to the class of “Brownian yet non-Gaussian diffusion”. Although this process results in the mean-squared displacement growing linearly with time, the distribution function is rather Laplacian than Gaussian for moderate time intervals and transit to the latter only when the range of random walkers’ displacements of random walkers overcomes the characteristic range of disordered inhomogeneities of the local diffusion coefficients. The reason of a high heterogeneity of local diffusion coefficients, respectively, to the transport of Gd-enhanced markers has also direct experimental evidence as reported in Ref. [86].

Remarkably, a quite similar picture was detected during simulations of the model of a particle’s motion in a random landscape of random diffusion coefficients slowly varying in space (quenched disorder) [69], see Figure 3A. Such evolution of the probability distribution function for a marker’s position, Laplace’s for short times and tending to Gaussian with the narrowing localised central peak (this feature, seen also in Figure 4B, was explored in detail in Ref. [85]) originates from the Gamma-distribution of the local diffusion coefficients (see Figure 3B). Note also that such behaviour was attributed to the Ito scheme of random walks with locally symmetric steps in space. Within this scheme, the spatial change of the diffusivity can is attributed to coordinate-dependent waiting times of a walker catches by randomly distributed traps [70] that looks reasonably from the point of view of complicated anatomy and topology of the brain’s extracellular space. This picture gains also recent attention for the simulation of a transport model in an explicitly compartmentalised medium, which comprises domains, where a particle can move freely, separated by rarely permeable corridors [83]. It is worth noting that the experimental work [87], where spatial locally correlated inhomogeneity was prepared by applying a special speckle pattern that allowed to track the transition from the normal diffusion of traced beads at very short times (within locally uniform patches) via a subdiffusion to the Brownian yet non-Gaussian diffusion at longer times.

### 4.3. SPT Results: Anomalous Diffusion of Transient Processes?

The movement of elongated nanoparticles was studied in [51] along and across their axis for MSD of individual tracks. While $MSD_{⊥}$ of perpendicular displacements is linear with time, the axial displacements $MSD_{\|}$ as a function of time typical for a subdiffusive process. This may indicate specific viscoelastic properties of the interstitial liquid retarding the random walks. However, this time interval is short and then $MSD_{⊥}+MSD_{\|}$ behaves linearly as should be for the normal Brownian motion. On the contrary, $MSD_{xy}$ determined for movements of the centre of mass in the laboratory frame is linear as a function of time during relatively short time intervals, approximately up to about 200 ms, see the dashed line in Figure 5 that allows to authors of [51] define this time interval as characteristic for exploration of local sub-domains and get spatial maps and distributions of local free volumes of ECS subdomains and coefficients of diffusion inside of them. The latter are determined as log-normal distributed.

At the same time, the quantitative properties of $MSD_{xy}$ for times t>200ms were discussed only qualitatively. The corresponding plot, redrawn in the double logarithmic coordinates, see Figure 5 indicates the typical subdiffusive behaviour with fractional power-index α<1. One can see that after a regime of normal diffusion acting when a nanoparticle walks within local intercellular cages, the transport slows down to subdiffusion with α=0.6 and further even more, to the case of α=0.5. Such a behaviour perfectly corresponds to the case of trapping within the frames of the continuous-time random walk (CTRW) model. Moreover, it is worth noting the Comb model, which describes such trapping of normally locally diffusing particles that leads to subdiffusion with α=0.5 in a direction orthogonal to local traps. The Comb model, which explains the emergence of subdiffusion with the power of the MSD index 1/2 from a normal diffusion, when a walker can be trapped in “teeth of combs” between displacements along it was proposed combinatorially for comb structure in the work [88] and later reformulated on in a language of diffusion equations [89]. The place of it and its generalizations place among the models of anomalous diffusion within the context of tracing individual trajectories is discussed in [90].

In the work [48], also operating with the single particle tracking of nanotubes in the ECS, the time range of tracks was extended up to decades of seconds. Two regions of the MSD separated by the interval looking as subdiffusion were revealed: short-time intervals, which allowed to plot the spatial distribution of instantaneous diffusion coefficients characterising transport properties within local cages separated by more narrow channels, and the asymptotic regime of large times (tenths of seconds) giving the global large-scale diffusion coefficient.

Remarkably, similar behaviour of the sequential change “normal diffusion”—“transient subdiffusion”—“asymptotic normal diffusion” was detected in another irregular porous medium, mucus [91]. The authors of the cited work also proposed the consider the time-varying MSD in the form
$$MSD\left(t\right)=\left(1-\frac{D_{0}}{D_{eff}}\right)MSD_{L}\left(t\right)+D_{eff}\left(t\right)$$
to cover both asymptotic regimes of the diffusion within local cavities of the size *L* and the global large scale/large time effective average diffusion, which can be detected in macroscopic observations; both diffusivities, $D_{0}$ and $D_{eff}$ can differ by orders.

Another example is the results of simulation studies for tracers spreading in polymer gels with irregular voids and tortuous paths [92]. The three-stage evolution of the MSD quite similar to Figure 5 and with the close value of α can be observed there. Note that kind of materials attracts recently attention as phantoms mimicking brain tissue [57].

It should be pointed out that the type of diffusive spread studied at the level of individual random walkers depends also on the characteristic size of the latter. In particular, the usage of quantum dots and quantum dots incorporated in small molecules of wheat germ agglutinin (smaller than nanotubes discussed above) indicates the MSD linear in time up to seconds that is typical for the normal Brownian motion [45]. However, the local diffusion coefficients are not uniform over the explored space in this case too, the probabilistic density distribution for them as well as for the local viscosities and local ECS dimensions are found. The dependence of the diffusion type on the coordination of walkers’ and paths’ seize is highlighted by the MDSs in ECS and within brain cells. In the second case, the subdiffusion originated from the complex tightly packed properties of intracellular components is detected already.

The spatial dimension of the walking space can also affect the value of the diffusion coefficient; recent development of super-resolution imaging, based, in particular, on the self-interference scheme allows 3D tracing of particles [93]; for the walking quantum dots, the Brownian type of diffusion is confirmed. Although the probability density functions for the diffusion coefficients along trajectories at different depths of slice differ to a certain extent, the averaged values for the 3D case are comparable with the value for 2D studies. This means that one can operate with measurements and simulations in thin slices when discussing the local diffusional properties.

## 5. Time-Dependent Diffusivity in the Physiological Conditions

Above, we showed that diffusion can have a non-classical character due to the properties of the propagation medium. However, a similar, if not greater, contribution to the change in the efficiency of molecular transport is made by a change in the proportion of ECS in the total volume of neural tissue, which occurs in response to changes in ionic concentrations, in particular, during the transition between sleep and wakefulness [9]. In recent years there have been more and more studies are emerging on changes in extracellular space volume affecting the transport of molecules, with changes in ES volume value varying according to the physiological state of the brain. A recent study showed a difference in the rate of diffusion of water during sleep and while awake [94,95]. It has been shown that the two components, slow and fast, of the apparent diffusion coefficient (ADC) change simultaneously with the volume of cerebrospinal fluid (CSF). Sleep versus wakefulness was associated with an increase in the slow component of ADC in the cerebellum and left temporal pole and a decrease in fast ADC in the thalamus, insula, parahippocampus, and striatum, and sleep awakening density was inversely associated with changes in ADC. CSF volume also increased during sleep and was associated with sleep-induced ADC changes in the cerebellum. There were no differences in ADC with wakefulness after sleep deprivation compared with wakefulness at rest. Thus results revealed both an increase in the slow part of ADC (mainly in the cerebellum) and a decrease in the fast part of ADC, which may reflect the different biological significance of fast and slow ADC values in relation to sleep.

Generally, a change in the volume of the extracellular space is associated with any change in the ionic composition of intercellular fluid during sleep and wakefulness, which is caused by transport processes through the membrane [96]. The transition from wakefulness to sleep has been shown to be accompanied by a marked and sustained change in the concentration of key extracellular ions and the volume of extracellular space. Arousal leads to a rapid increase in potassium with a simultaneous decrease in calcium, magnesium and protons and a decrease in extracellular space. Normal sleep or anaesthesia causes the opposite change in extracellular ion concentration and is accompanied by an increase in extracellular space volume. At the same time, changes in potassium concentration in both cases occurred within a few seconds, while the dynamics of changes in calcium and magnesium were rather slow in the transition from sleep to wakefulness. It was also shown that the differences in concentrations of key ions were maintained in different states for quite a long time in freely behaving animals.

Besides sleep-wake transition, it has been shown that ECS volume changes dynamically in response to physiological and pathological neuronal activity and that these changes have important functional consequences [27,97,98,99,100]. In this case, characteristics such as diffusion and volume changes can be markers of brain conditions. When the volume of the ECS decreases—neurotransmitters and neuromodulators affect a large population of cells, increasing the extra-synaptic and volumetric connectivity between brain cells, which can lead to serious disturbances. In particular, a recent review [98] on the relationship between ECS changes and seizures provides evidence that sudden or prolonged shrinking of the ECS may create the conditions for seizure initiation, as well as contribute to seizure maintenance. This is because ECS volume is one of the key factors affecting the increase/decrease of neuroactive substances such as ions and neurotransmitters and an increase in ephaptic interactions. For example, a reduction in ECS can simultaneously concentrate surrounding glutamate and K+ and cause a greater overlap of electric fields of neighbouring neurons. All these cascading changes will both promote greater neuronal excitation and, if the contraction occurs over a large area at the same time, will promote synchronous discharge, which can lead to a seizure. The mechanisms that can lead to such a pathological condition are ultimately controlled by the ECS, such as electrolyte imbalance in the ECS or increased non-synaptic/ephaptic interactions between neurons.

A study [99] of epileptic seizures in a mouse experimental model has shown that the extracellular space volume is reduced by almost 15% in vivo. Different pharmacological blockers were used to eliminate epileptic activity and stop the reduction of the ECS volume. The authors argue that the results obtained in vitro and in vivo, about the relationship between epileptic seizures and changes in ECS volume allows to target research on inhibition of changes in ECS volume, which will help in the future treatment of patients with epilepsy resistant to current treatments.

It should be noted that viscoelasticity of the extracellular matrix should not be excluded from a role in changes in ECS volume. An extensive review [101] of numerous studies in the past two decades of the extracellular matrix discusses its overall influence on fundamental cellular processes, including proliferation, growth, proliferation, migration, differentiation and organoid formation. From this perspective, the elasticity and stiffness of the matrix are worth considering when assessing the qualitatively different types of the spread’s behaviour in the brain. In addition, we can also mention the recent work [102], where the effects of viscoelasticity of biomimetic matrices are discussed in relation to properties of biological tissues in general, applications as scaffolds for cellular cultures and for physical models of related structure-transport interrelations in general.

## 6. Conclusions

Since our article has a dual character, the Conclusions are also addressed to researchers in two different fields.

To biologists and physiologists who study the living brain, we address the above information that diffusion processes are by no means always characterized by Gaussian statistics and obey Fick’s law. This means that when statistically processing experimental data, it makes sense, for example, to compare the approximations of the Gaussian and exponential functions, and when analyzing the transport of substances in brain tissues, at least keep in mind alternatives from the Brownian yet non-Gaussian diffusion area.

In turn, we would like to draw the attention of physicists studying stochastic processes to experimental studies of the processes of substance transport in brain tissues. Nicholson’s with colleagues long series of work on diffusion in the brain, combined with the recent (and unfinished) scientific discussion about the mechanisms of drug transport in the parenchyma, provide a good basis for developing advanced theoretical models of diffusion itself. Namely, the combination of the complex shape of the intercellular space, the presence of traps (dead spaces) and obstacles in the form of large molecules, as well as the dynamic regulation of this volume, for example, during the transition between sleep and wakefulness—all these properties deserve theoretical and model studies, the main result which it may be possible to determine the features of the structure of the intercellular space of the brain parenchyma by the measured characteristics of the transport of molecules in it.

## Figures and Tables

**Figure 1 ijms-23-12401-f001:**
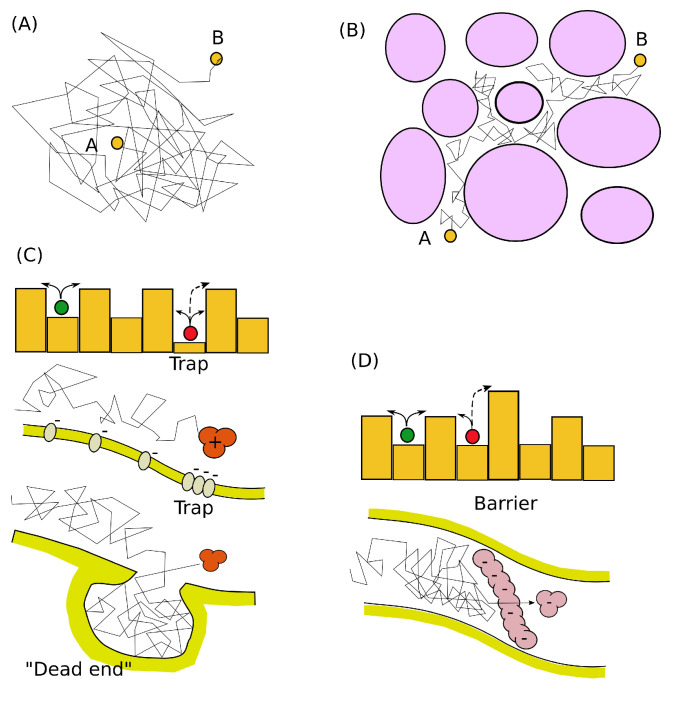
(**A**) The free diffusion process represented by a single particle trek (random walk) from point A to point B; (**B**) The random walk through obstacles can be treated as free diffusion over a longer distance using effective diffusion coefficient and tortuous parameter; (**C**) The theoretical model of trapping implies the existence of low-potential sites which are difficult for a particle to escape. It may be represented either by the locally charged molecules or by ”dead ends” of extracellular space; (**D**) The theoretical model of the barrier assumes the presence of a region that requires high energy to overcome it. In extracellular space, it may have the form of large charged molecules of the extracellular matrix.

**Figure 2 ijms-23-12401-f002:**
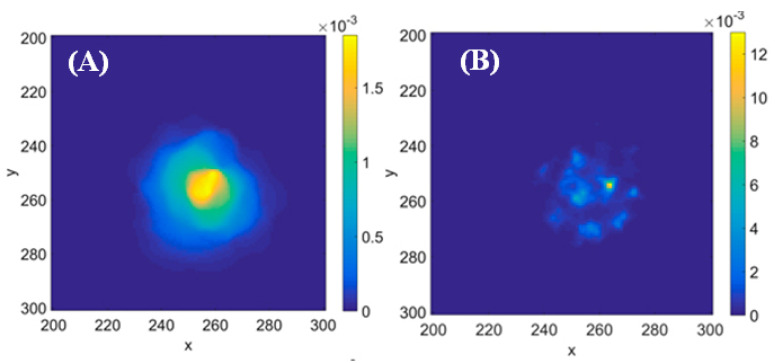
Examples of concentration distributions of a marker spreading from a single point within Hänggi–Klimontovich’s (**A**) and Ito’s (**B**) realisations of the random walk in the medium with Gamma-distributed and then locally correlated (with the correlation length ℓ=10 lattice sites) local diffusivity. Reproduced from [69] (published there under Creative Commons Attribution 4.0 licence).

**Figure 3 ijms-23-12401-f003:**
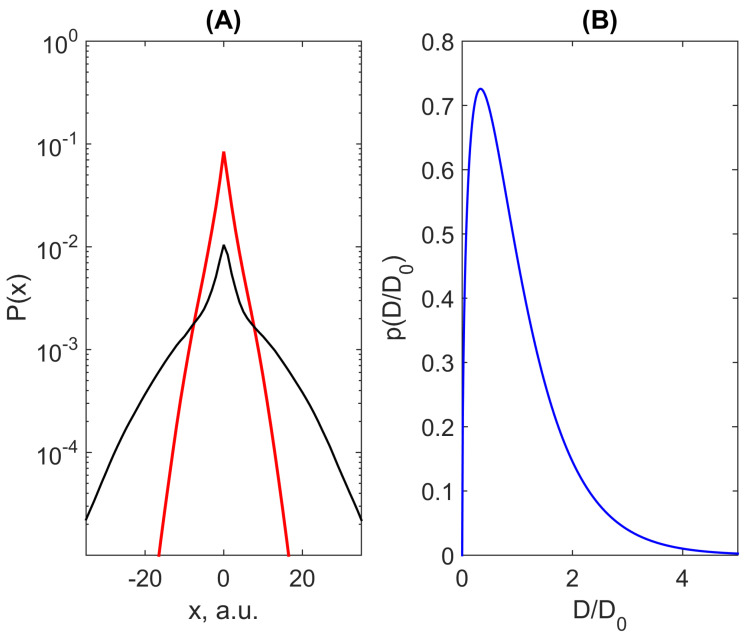
(**A**) Examples of evolution of probability density function for the Brownian yet non-Gaussian diffusion in 2D heterogeneous media simulated accordingly to the equilibrated Ito scheme for two time moments (t=10, red line, and t=100, black line) adapted from [69] (published there under Creative Commons Attribution 4.0 licence) (**A**) and the Gamma-distribution of diffusion coefficients mimicking the heterogeneity of the medium, respectively, to the uniform value $D_{0}$ (**B**).

**Figure 4 ijms-23-12401-f004:**
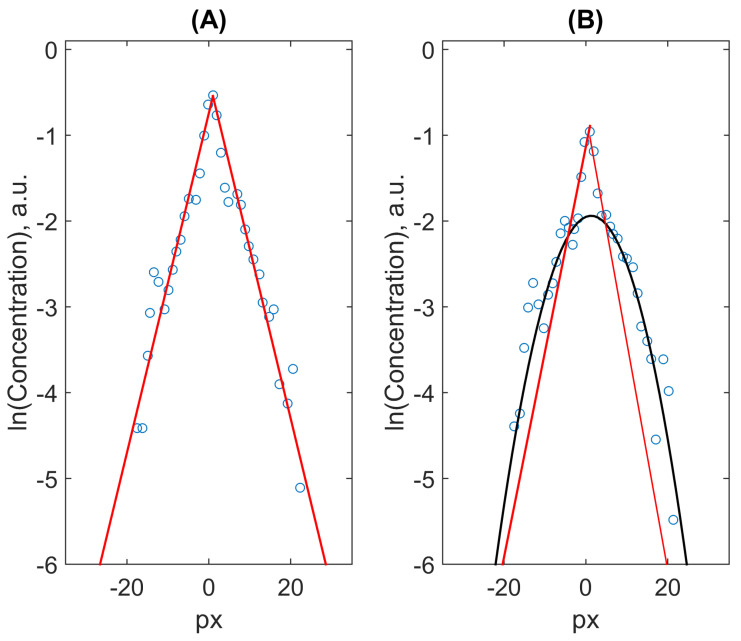
Examples of digitized data (markers) of two distributions taken from [39], for 59s (**A**) and 72s (**B**) and their fits by the Laplace (red lines) and the Gaussian (black parabola) functions.

**Figure 5 ijms-23-12401-f005:**
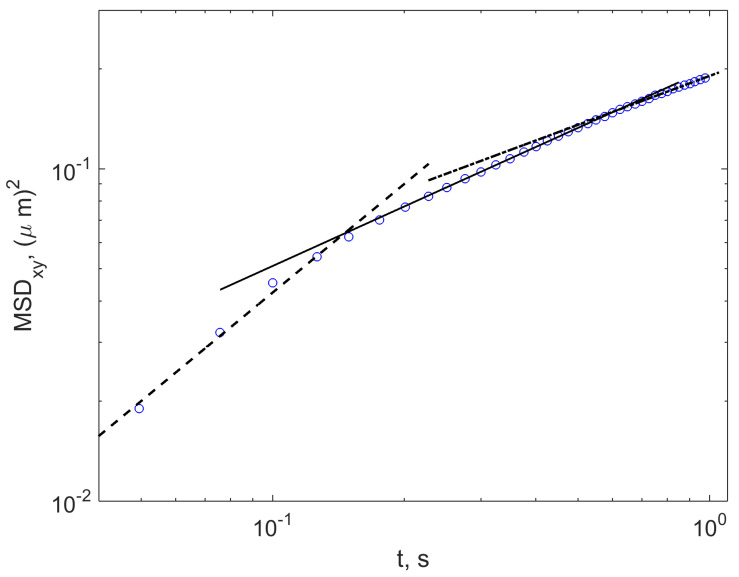
The digitized data for 2D movements in the laboratory frame (markers) taken from [51], digitized and replotted in double logarithmic co-ordinates. The straight lines indicated the power-law dependencies $MSD_{xy}∼t^{\alpha }$ with α=1 (dashed line), α=0.6 (solid line), and α=0.5 (dash-dotted line).

## Data Availability

Not applicable.

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
