# Peer review of "Transport in the Brain Extracellular Space: Diffusion, but Which Kind?"

_ijms, 2022, doi:10.3390/ijms232012401_

Round 1

Reviewer 1 Report

The current review aims to summarize the available knowledge on the transport of substances in the brain parenchyma, with a focus on the two experimental techniques (MRI and SPT or single particle tracking), which have potential to provide data of a special interest related to non-conventional kinds of diffusion.

The review's structure is well-designed and the manuscript is well-written. However, it is suggested that the writing can be subjected to improvements and checking for typo errors. As examples, there are few minor points and typo errors which need to be addressed by the authors: 

- Line 87: "depends of the the membrane permeability" is suggested to be corrected.

- Line 88: "All this complicate" is suggested to be rephrased.

- Line 93: "permit" should be permeate?

- Line 99: "contract agents" should be contrast agents?

- Line 105: it is suggested that the authors check to be sure if the "labirintine" is written correctly.

- Line 109: do authors mean SPT approach by "SRT approach"?

- Line 140: "particualar" should be re-written.

- Line 142: "we will considered" should be re-written.

- Line 144: "In many cases, these two approaches do not contradict to each other. For the usual diffusion ( which is ergodic process) one can characterize it with single diffusion coefficient D,either typical for a medium or an effective one" it seems that a point "." is missing at the end of the sentence. 

- Line 153: "inhomogenety" should be checked for spelling errors.

- Line 202: "the authors lists" is suggested to be cheched for grammatical error.

- Line 223: do authors mean ECS by "ESC"?

Author Response

 Dear Reviewer,

We are grateful  for the positive assessment of the work and valuable suggestions. Please find below our response to comments. 

.. there are few minor points and typo errors which need to be addressed by the authors: ... 

Answer: 

All mentioned typos are corrected (not highlighted in color). We have proofread all the text  in order to catch and correct both mentioned and   more typos. In revised manuscript, all corrections made are highlighted in colors.

Reviewer 2 Report

First of all, it must be said that the topic of this review is not entirely my area of expertise. 

In any case I would recommend to authors to carefully reread the manuscript. There is a significant number of errors in the work - typos, duplicated words, missing verbs, wrong tenses.. The revision of the manuscript by a native speaker would greatly enhance the readability of the text.

Some abbreviations are wrongly established (too late, twice) or misspelled (ESC/ECS, SPT/SRT) and the abbrev list is incomplete.

The review tries to appear multidisciplinary. Therefore, some terms commonly used in the field could be explained for the readers from other disciplines (e.g. walker). 

From my point of view, the third chapter could be supplemented with simple schematic pictures of the described phenomenoms (particle trajectories, trapping,..).

The biggest question for me is the added value of this work. Many other up-to-date articles and reviews are cited, which is good. But given their existence, I wonder why the authors decided to write this review?

Last but not least, due to the strong physics orientation of this work, I would recommend considering publication in another journal. 

Author Response

Dear Reviewer,

We are grateful  for the positive assessment of the work and valuable suggestions. Please find below our response to comments. In revised manuscript, all corrections made are highlighted in colors.

Comment 1:In any case I would recommend to authors to carefully reread the manuscript. There is a significant number of errors in the work - typos, duplicated words, missing verbs, wrong tenses.. The revision of the manuscript by a native speaker would greatly enhance the readability of the text.

Answer:  We accept this recommendation. We have checked  all the text through the manuscript and made multiple corrections. 

Comment 2: Some abbreviations are wrongly established (too late, twice) or misspelled (ESC/ECS, SPT/SRT) and the abbrev list is incomplete.

Answer: We have checked  and corrected all the abbreviations used. The  abbreviations list was revised and updated.

Comment 3: The review tries to appear multidisciplinary. Therefore, some terms commonly used in the field could be explained for the readers from other disciplines (e.g. walker). 

Answer:  We agree with this comment. We have introduced explanations of key terms from stochastic theory, specifically :  “ergodicity” (line 123),  “walker” ( line 204).

Comment 4: From my point of view, the third chapter could be supplemented with simple schematic pictures of the described phenomenoms (particle trajectories, trapping,..).

Answer: It has been done. Figure 1  in the revised manuscript illustrates the  free  diffusion as random walk, the random walk in  tortuous medium, the particle trapping, and the barriers.

Comment 5: Many other up-to-date articles and reviews are cited, which is good. But given their existence, I wonder why the authors decided to write this review?

Answer: You are right, there are a number of good reviews on both fluid transport in the brain and on modern developments in the field of random walks and non-classical diffusion, they are cited. However, the main drawback of these reviews is that their authors consider these problems separately, exclusively within their area of interest, either biological or physical. As a result, works devoted to studying diffusion in the brain's extracellular space operated exclusively with classical picture of Brownian motion and (Gausian, Fickian) diffusion although there is evidence that such simple models may be not adequate to the picture registered in the considered complex medium. On the other hand, physicists reviewing recent advances in the theory of anomalous diffusion are limited to either simulational examples or examples related to intracellular motion on motion of proteins on a cell's membrane; recent results related to the transport in the brain's parenchyma are overlooked. 

Thus, our primary goal is to try to bring the two communities together, highlighting those experimental results in studying ECS, which devote interest to physicists to the advanced modelling as well as to give an overview of specific physical models developed recently which may find their application to the transport phenomena in the brain. For this reason, the topics of this review are primarily focused on two directions, which look prospective within the stated goal (Brownian-yet-not-Gaussian diffusion and Single-Particle Tracking), the rest is mentioned shortly and referenced to existing reviews. At the same time, for two mentioned topics, we not only simply listed the published papers but also replotted some of the published data in the form, which highlights the discussed open questions and follows the goal of attracting attention of  the multidisciplinary community. 

The line of reasoning explained above, is formulated as “ a dual character” of this work and addressed to both physicists and biologists drawing attention of both communities to specific problems, which are more studied by one of them and deserve attention from another.

Comment 6: Last but not least, due to the strong physics orientation of this work, I would recommend considering publication in another journal. 

Answer: We are grateful to the referee for this suggestion. However, our intention is to address  the interdisciplinary audience that IJMS readers form. For an article addressed more to physicists, we see a different structure, with a consistent presentation of the structural features and properties of the extracellular space as a medium of physical processes.

Reviewer 3 Report

The authors present a review on the nature of transport phenomena in the brain. The manuscript is generally well written, but I suggests to include comparative tables to better outline the different studies and diffusion models.

In order to improve the application of the notion presented in the paper I also suggest to include some direct consideration with experimental or clinical cases. For instance, it is know that mechanical and transport properties are intrinsecaly coupled in soft tissues such as the brain (https://www.liebertpub.com/doi/full/10.1089/ten.teb.2021.0151) . Therefore the authors may better discuss this aspect, highlighting how these properties can change during ageing or in pathological conditions.

Author Response

 Dear Reviewer,

We are grateful  for the positive assessment of the work and valuable suggestions. Please find below our response to comments. In revised manuscript, all corrections made are highlighted in colors.

Comment 1: In order to improve the application of the notion presented in the paper I also suggest to include some direct consideration with experimental or clinical cases. For instance, it is know that mechanical and transport properties are intrinsecaly coupled in soft tissues such as the brain (https://www.liebertpub.com/doi/full/10.1089/ten.teb.2021.0151) . Therefore the authors may better discuss this aspect, highlighting how these properties can change during ageing or in pathological conditions.

Answer: We generally accept this comment, but would like to note that a detailed treatment of these issues would be sidetracking in our manuscript. It seems to us that the questions of statistical characteristics and models of transport in brain tissues are currently aimed at understanding some experimental data, while their evolution with age and pathologies is a serious subject of future research.

In response to this comment  we have included a brief discussion of the specific issue mentioned by the reviewer in section 5 with an appropriate reference.